# Genome-Wide Association Mapping for Heat Stress Responsive Traits in Field Pea

**DOI:** 10.3390/ijms21062043

**Published:** 2020-03-17

**Authors:** Endale G. Tafesse, Krishna K. Gali, V.B. Reddy Lachagari, Rosalind Bueckert, Thomas D. Warkentin

**Affiliations:** 1Department of Plant Sciences, College of Agriculture and Bio-resources, University of Saskatchewan, Saskatoon, SK S7N 5A8, Canada; endale.tafesse@usask.ca (E.G.T.); kishore.gali@usask.ca (K.K.G.); rosalind.bueckert@usask.ca (R.B.); 2AgriGenome Labs Pvt. Ltd., Hyderabad 500 078, India; vb.reddy@aggenome.com

**Keywords:** pea, heat stress, genetic diversity, GWAS, genotyping-by-sequencing, marker-trait association, candidate-gene

## Abstract

Environmental stress hampers pea productivity. To understand the genetic basis of heat resistance, a genome-wide association study (GWAS) was conducted on six stress responsive traits of physiological and agronomic importance in pea, with an objective to identify the genetic loci associated with these traits. One hundred and thirty-five genetically diverse pea accessions from major pea growing areas of the world were phenotyped in field trials across five environments, under generally ambient (control) and heat stress conditions. Statistical analysis of phenotype indicated significant effects of genotype (G), environment (E), and G × E interaction for all traits. A total of 16,877 known high-quality SNPs were used for association analysis to determine marker-trait associations (MTA). We identified 32 MTAs that were consistent in at least three environments for association with the traits of stress resistance: six for chlorophyll concentration measured by a soil plant analysis development meter; two each for photochemical reflectance index and canopy temperature; seven for reproductive stem length; six for internode length; and nine for pod number. Forty-eight candidate genes were identified within 15 kb distance of these markers. The identified markers and candidate genes have potential for marker-assisted selection towards the development of heat resistant pea cultivars.

## 1. Introduction

Pea (*Pisum sativum* L., 2*n* = 14) is a major pulse crop widely grown in the temperate regions primarily for its nutritional values as a source of protein, slowly digestible starch, essential minerals, high fiber and low fat; and soil fertility benefits as it fixes atmospheric nitrogen [1,2,3]. However, as a cool season crop, pea is prone to heat and drought stress, with warm summers causing shortened life cycles, abortion of floral components and pods, and thus economic yield loss [4,5,6]. Due to global warming, the average surface temperature is predicted to increase by 3.7 °C by the end of this century, and thus heat stress is expected to be even more challenging in the future [7].

Genetic improvement of pea for heat and drought resistance is a promising approach to stabilize yield under environmental stresses. Pea germplasm has a wide range of diversity in morpho-anatomical, biochemical and physiological characteristics [8,9]. Among other things, such diversity has been explored to identify traits associated with heat response [10,11,12]. Pigments including chlorophylls, carotenoids, anthocyanins contribute to heat tolerance through heat dissipation and protection of vital plant components and processes [13,14]. Multi-environment studies on pea [10], and maize [15] revealed leaf color (greenness) as a trait linked to stress tolerance.

Chlorophyll represents pigment abundance and composition, and is used to drive photosynthesis, plant senescence, and yield potential [15,16]. Stay-green, a trait that delays plant senescence, is reported to be associated with improved yield under stress conditions [15]. Estimation of leaf chlorophyll concentration by the soil plant analysis development (SPAD) meter is reliable, and is strongly correlated with laboratory-based destructive methods [17].

Vegetative indices (VI), determined from different wavelengths of spectral reflectance, have been used as proxies to quantitatively and qualitatively assess traits linked with vegetation cover and plant vigor, pigment abundance and composition, and plant water status [18,19]. Thus, VIs indirectly indicate the overall physiological state of the plant under various environmental conditions. For example, photochemical reflectance index (PRI), derived from narrowband wavelengths, indicates photosynthetic efficiency and photosynthetic performance in stress [19]. Canopy temperature (CT) is a direct indicator of degree of stress in plants. If CT is greater than the air temperature, then the plants are under stress predominantly caused by heat and drought. Although the environment contributes to CT to a great extent, there exists significant variation in genotype response [12].

In pea and other crops, lodging is one of the plant factors that exacerbates heat stress by making the plant hold more heat in the canopy, and thereby leading to increased CT [12,20]. Heat and drought stress decreases reproductive stem and internode lengths [12], which are related to genes associated with gibberellin function [21,22]. Pod number, a major yield component in pea and other pulse crops, is an economic trait highly affected by heat stress [23,24]. Pod loss due to heat stress is mostly associated with pollen and stigma malfunction, and abortion of flowers, bud and pods [6,11].

Understanding of the genetic base of traits involved in pea stress response would assist breeders in developing heat resistant varieties. Genome-wide association study (GWAS) has been used as a tool for dissecting the genetic bases of various traits using the naturally occurring genetic diversity a species has accumulated over many generations [25,26]. Linkage disequilibrium (LD)-based association mapping provides high resolution, as it relies on the use of single nucleotide polymorphisms (SNP), and thus has the capacity to distinguish even between closely related individuals [27,28,29,30]. The advancement and inexpensive availability of high-throughput next generation sequencing (NGS) platforms enabled the use of SNPs for genetic diversity study and estimation of LD in pea and other crops [29,30]. Association mapping has been successfully used for identification of numerous genomic loci and underlying genes for complex traits in several crops including pea [25,26,27,28,29,30,31,32,33,34,35].

In pea, association and linkage mapping has been employed to uncover the genetic bases of several traits including agronomic and seed quality traits [30,35], disease resistance [32,36], seed mineral concentrations [37], seed lipid content [38], salinity tolerance [31], and frost tolerance [33]. Despite its importance, only limited studies have been carried out to identify genomic regions associated with pea stress tolerance [28]. Stress tolerance is complex and is controlled by many genes throughout the genome each with minor effects and each interacting with the environment [39]. The objectives of this study were to examine the G × E interaction in pigment and vegetative structures associated with stress response, to explore the genetic variation of stress tolerance present in a GWAS panel of 135 accessions, and to identify MTAs related with six stress responsive traits.

## 2. Results

### 2.1. Weather and Stress Condition of the Environments

The weather condition of the five environments during the pea growing season described by the average of daily maximum, minimum, 24 h mean temperatures, number of days when the daily maximum temperature was greater than 28 °C during the growing season, and total monthly precipitation is summarized in Table 1. In pea, significant yield loss due to heat stress is evident whenever the daily maximum air temperature exceeds 28 °C for several days during the growing season [5]. Impact of heat and drought is severe when it occurs during reproductive stages. Saskatoon 2015 was the most stressed environment as indicated by mean daily maximum air temperatures > 27 °C, 18 days where air temperature was > 28 °C, and drier conditions during the reproductive stage. Similarly, 2017 Saskatoon was also under heat and drought stress during the reproductive stage with average air temperature ~26 °C, 16 days where air temperature was > 28 °C, and relatively low total precipitation. The remaining three environments were generally ambient and considered as control environments (Table 1).

### 2.2. Phenotypic Measurements, Analysis of Variance, and Marker Detection through Association Mapping

Variance components of genotype (G), environment (E), and G × E interaction together with their significance on the six traits used in this study is presented in Table 2. For all traits analyzed, normality of residuals and homogeneity of variance were met.

Descriptive statistics for minimum, maximum and mean values of phenotypic measurements on the traits of the GWAS panel across five environments are summarized in Table 3 and Figure 1.

Chlorophyll concentration, measured by a SPAD meter, was affected by genotype, environment and their interaction; and the variance component analysis showed that maximum variation (67.9%) among the GWAS panel was due to the genotype effect, and the broad sense heritability was 0.95. Overall, genotype chlorophyll concentration ranged from 26.6 to 57.6 SPAD values under heat stress, and 30.0 to 67.5 under control conditions (Table 3). On average, the heat stressed environments had 3% less SPAD value than the ambient environments. Six markers (Chr5LG3_150942510, Chr5LG3_446272814, Chr5LG3_449362407, Chr5LG3_566189589, Chr5LG3_569788697, and Chr5LG3_572899434) were associated with SPAD in at least three out of the five environments, and on average each marker explained 7%–13% of the phenotypic variance (PV) measured as the difference in R-square of the model with the SNP and without the SNP. SNP markers Chr5LG3_566189589 and Chr5LG3_449362407 were associated with SPAD in 4 and 5 environments explaining 13% and 11% of the PVs, respectively (Table 4). PRI was also significantly affected by genotype, environment and by the G x E interaction. Variance components showed most of the variation in PRI was due to environmental factors, and the broad sense heritability was the least (0.35) compared with the other traits (Table 2). Two markers, Chr6LG2_469101917, and Chr7LG7_263964018 were significantly associated with PRI at three out of the five environments. Each of the two markers explained 9% of PV (Table 4). 

For canopy temperature (CT), the GWAS accessions significantly varied due to both genotype (G) and environment (E) effects, but not by the G x E interaction (Table 2). In general, under heat stress, the accessions’ CT, measured four to six times in a season during reproductive stage on hot days at solar noon, ranged from 24.5 to 31.0 °C, whereas under ambient conditions, the CT ranged from 21.4 to 26.9 °C. This temperature difference indicated that CT is highly influenced by the environment effects with a relatively lower broad sense heritability of 0.57 (Table 2; Table 3; Figure 1). Two SNP markers (Chr4LG4_415994869 and Chr5LG3_309595819) were associated with CT in three of the five environments. The R-square value of the model with SNP ranged from 0.43 to 0.53, and each of the SNP markers explained 6% of PV. 

Reproductive stem length was also affected by genotype and environment main effects and their interaction. The reproductive stem length under the stressed environments ranged from 13 to 99 cm, whereas under the control environments the range was from 14 to 117 cm, suggesting heat stress decreased the reproductive stem length. Analysis of variance components showed genotype and environment main effects respectively contributed to 63.4% and 7.6% of the variation in the GWAS panel. The broad sense heritability for reproductive stem length was 0.92. Seven SNP markers (Chr3LG5_18678117, Chr4LG4_29062302, Chr5LG3_566189271, Chr5LG3_572669963, Chr7LG7_20086906, Chr7LG7_23295474, and Chr7LG7_96157380) were associated with reproductive stem length in at least three of the five environments, and four of these SNPs were consistent in at least four of the five environments. SNP marker Chr4LG4_29062302 was found to be associated with the trait in all five environments with an average R-square of the model of 0.60. Overall, the R-square value of the model with SNP ranged up to 0.71 for reproductive stem length (Table 4).

Internode length was another trait significantly affected by genotype and environment main effects and their interaction. Under heat stress, the internode length ranged from 1.6 to 11.3 cm with a mean value of 11.0 cm, whereas under control conditions, the range was 1.9 to 14.9 cm with a mean value of 14.8 cm. Variance component analysis showed genotype and environment respectively contributed 43% and 4.8% of the variations to the GWAS panel. The broad sense heritability was 0.90. Six SNP markers (Chr4LG4_62461234, Chr4LG4_63111072, Chr4LG4_80759704, Chr5LG3_566189271, Chr6LG2_420562729, and Chr7LG7_197862543) were associated with internode length in at least three of the five environments. These markers were significantly associated with internode length in at least three of the five environments with the R-square value of the model with SNP ranged up to 0.63. SNP marker Chr5LG3_566189271 was identified in all five environments with an average R-square of 0.49.

Pod number was also significantly affected by genotype and environment main effects and their interaction. Variance component analysis showed genotype and environment, respectively, contributed 36.6% and 12.4% to the overall pod number variance in the GWAS panel. Compared with the three control environments, pod number under the heat stress environments decreased by 14.6%. The broad sense heritability in pod number was 0.88. Eight SNP markers (Chr2LG1_4359822, Chr2LG1_105547608, Chr2LG1_370541780, Chr2LG1_385949935, Chr2LG1_389336188, Chr2LG1_402022079, Chr3LG5_216337201, Chr5LG3_530537682, and Sc04062_32372) were associated with pod number in at least three of the five environments explaining 7% to 9% of PV, with an average R-square value of 21.9.

Manhattan plots showing the association of SNP markers with plant chlorophyll concentration and reproductive stem length in multiple trials, and the corresponding Q-Q plots are presented as examples from this research in Figure 2 and Figure 3, respectively. The Q-Q plots represent the observed P values of each SNP marker against the expected P values. The Manhattan plots in Figure 2 showed the significant association of SNP markers on Chr 5 (LG3) with plant SPAD in each of the individual environments presented. The Manhattan plots in Figure 3 showed the significant association of SNP markers on multiple chromosomes with the reproductive stem length. 

Of all the MTAs that were observed in > 60% of the environments, the following markers had the greatest percent variation averaged over the selected environments for the respective traits: Chr5LG3_566189589 (13% PV) and Chr5LG3_449362407 (11% PV) for SPAD; Chr6LG2_469101917 and Chr7LG7_263964018 each with 9% PV for PRI; Chr4LG4_415994869 and Chr5LG3_309595819 each with 6% PV for CT; Chr3LG5_18678117 (6% PV), Chr5LG3_572669963 (5% PV), and Chr7LG7_96157380 (5% PV) for reproductive stem length; Chr4LG4_63111072 (6% PV), Chr5LG3_566189271 (7% PV) and Chr4LG4_62461234 (6% PV) for internode length; and seven markers, Chr2LG1_105547608, Chr2LG1_370541780, Chr2LG1_385949935, Chr2LG1_389336188, Chr3LG5_216337201, Chr5LG3_530537682, and Sc04062_32372 each with 9% PV for pod number (Table 4).

Forty-eight unique genes were identified within a 15 kb region of the selected 32 SNP markers and are considered as candidate genes. The candidate genes identified for various traits included those encoding for transcription factor, translation initiation factor, heat shock protein, ribosomal protein, protein kinase, transmembrane protein, and others as listed in Table 5. Two genes, Psat5g299080 and Psat5g299040, which encode the proteins kinesin-related protein 4-like and PPR containing plant-like protein (putative tetratricopeptide-like helical domain-containing protein), were identified as potential candidate genes associated with internode length, reproductive stem length and chlorophyll content (SPAD).

### 2.3. Overall Association of Phenotypic Traits

Principal component analysis (PCA) based on the correlation of traits revealed the overall traits association and the genotype response across the five environments (Figure 4A,B). The first two PCs explained 61.9% of the total variability in the data. The loading plot illustrated traits association and how much each trait contributed to the PCs. The first PC was influenced mainly by SPAD, reproductive stem and internode lengths, whereas the second PC was influenced mainly by CT and pod number. SPAD positioned in an opposing direction (obtuse angle to straight line) to reproductive stem and internode lengths indicating a significant negative correlation between SPAD and the length measurements. Likewise, CT positioned in the opposite direction of pod number indicating their significant negative correlation. The hotter the canopy, the lower the pod number and thus seed yield (Figure 4A). Score plots illustrated genotype placement (response) across the environments (Figure 4B). The heat and or drought stressed environments (2015 Saskatoon and 2017 Saskatoon) positioned to the negative direction PC2 associating with high CT, whereas the control environments were associated greater pod number and SPAD value. 

## 3. Discussion

As a cool season crop, pea is sensitive to heat stress which causes a significant yield loss. However, there exists substantial genetic variation among pea genotypes for heat tolerance [10,12,24,28]. A strategic assessment and use of available variation is essential for crop improvement through using allelic variation. With the availability of cost-effective, high-throughput SNP genotyping methods and genomic resources, GWAS has been an effective method for identifying genetic loci associated with traits of many crop species including legumes [29,30,36]. 

The present GWAS was undertaken to identify SNP markers associated with traits related with pea heat response using a panel of 135 genetically diverse pea accessions. The accessions were from breeding programs of major pea growing areas and, thus accounted genotypes with a wide range of heat sensitivity. Genotyping by sequencing (GBS) identified 16,877 good quality SNPs, of which 15,609 were distributed across seven chromosomes of pea and the remaining 1268 were non-chromosomal SNPs [30]. 

Linkage disequilibrium patterns of population structure and genetic relatedness information are important for association mapping to minimize the number of false positive associations [41], thus the LD of the 135 GWAS members was analyzed by chromosome, and the LD decay estimates of the 7 chromosomes ranged from 0.03 to 0.18 Mb [30]. Based on genetic relatedness the 16,877 SNPs in the GWAS panel were clustered into nine groups [30]. Similarly, Diapari et al. [37] clustered another 94 pea accessions into eight groups, and Siol et al. [42] grouped 917 *Pisum* accessions into 16 groups. The above groupings indicated the extent of genetic variability among pea accessions. The clustering did not necessarily correspond solely with the geographic origins of the individuals, but depended on additional factors of variability such as the objectives in different breeding programs [30]. 

In the present GWAS, we evaluated ten heat stress-responsive traits. The first six were: chlorophyll concentration by SPAD, PRI, CT, reproductive stem length, internode length, and pod number. The other four were: plant height, lodging, pod to node ratio, and water band index (WBI). From the latter four traits, five SNP markers on Chr 1 (LG6), Chr 2 (LG1), Chr 3 (LG5), Chr 5 (LG3) for lodging, and four SNP markers on Chr5 (LG3) for plant height were previously reported by Gali et al. [30], and no marker was detected to be significant in at least three of the five environments for pod to node ratio and WBI. As such, in the current paper we focused on the first six traits for phenotypic variation in the 135 pea accessions across five environments. 

The five environments were grouped into ambient (three environments) and heat and or drought stress (two environments) conditions based on weather data and threshold temperature for heat stress in the field [5]. All traits had a wide range of phenotypic variation within each environment and stress level, which is essential for dissecting complex traits through association mapping. Overall, we identified 32 MTAs for six traits that have physiological and agronomic importance and are involved in pea heat response. A marker identified for a significant association with a given trait would be more reliable if the same marker is found in multiple environments [30]. Therefore, for the six traits we investigated, the SNP markers deemed significant were consistent in at least three environments, and these markers could potentially be used for marker-assisted selection of these traits in the effort of improving pea for heat tolerance. 

In this study, the SPAD value was used to estimate chlorophyll concentration, a major component of chloroplasts, and can be used as a factor to determine crop adaptation to environmental stresses by retention of greenness [10,13,43]. Regression analysis on wheat reported that under heat stress, the SPAD value was associated with plasma and thylakoid membrane damage [44], which hinders light absorbing efficiency of photosystems (PSI and PSII), and hence reduced photosynthetic capacity ultimately leading to crop yield loss [11]. Understanding of the genetic bases that govern chlorophyll concentration may contribute to enhancing photosynthetic efficiency and thus minimize yield loss due to stressful environments. 

We identified six MTAs that were related to SPAD value in repeated tests. All of the MTAs identified for SPAD were from Chr 5 (LG3). Bell et al. [45] reported that pea chlorophyll degradation under stress conditions is governed by the SGRL protein, a distinct class of the SGR gene which is induced by environmental stresses. The genomic location of SGRL was reported to be on LG3 which supports our result where all of the SPAD markers also reside on Chr 5 (LG3). The SGRL gene sequence (https://www.ncbi.nlm.nih.gov/nuccore/LN810021) location in the pea genome assembly spanned between the base pair positions Chr5LG3_151800929 and Chr5LG3_151804253, and is within close proximity (858 Kbp) of the SPAD associated marker ChrLG3_150942510. Using GWAS on soybean, Dhanapal et al. [29] identified 52 SNP markers associated with chlorophyll content.

Similarly, two loci were identified for association with PRI. One of these loci is on Chr 6 (LG2) and the second is on Chr 7 (LG7), and in both cases the markers were consistent in three environments. There are only a few reports that have used GWAS to identify markers associated with vegetation indices, namely, in soybean and wheat. In soybean, Herritt et al. [25] identified 31 SNPs linked with PRI, and on wheat, Gizaw et al. [34] reported the presence of markers associated with normalized chlorophyll-pigment ratio index (NCPI), and normalized difference vegetation index (NDVI). However, use of GWAS and vegetation indices has been lacking in cool season pulse crops. To the best of our knowledge, our report is the first to apply VIs in pea GWAS. The PRI is increasingly used as a predictor of crop photosynthetic efficiency which responds to environmental variables [19]. PRI is associated with photosynthetic protective mechanisms by dissipation of excess energy such as in the operation of xanthophyll cycle during stress. Violaxanthin de-epoxidase VDE is among the genes known to be involved in excess energy dissipation in the xanthophyll cycle [46].

Two MTAs, one each on Chr 5 (LG3) and Chr 4 (LG4), were detected for CT, a trait consistently used as an indicator of stress mainly of drought and heat stresses [12,14]. Generally, cooler canopy is associated with heat avoidance, and is an indicator of a healthy canopy with an optimal physiological state [12]. Again, to the best of our knowledge, no previous study exists on pea or other cool season legume crops that has reported genomic regions associated with CT. In a study using 24 pea cultivars across six environments, Tafesse [14] reported that leaf surface wax concentration is positively correlated with water band index, a proxy for leaf water retention, and contributes to a cooler canopy. WAX2 is among the genes controlling wax biosynthesis in Arabidopsis [47], and glossy13 is another gene with similar role reported in maize [48]. Lodging contributes to canopy heating in pea, and upright and semileafless cultivars with the *afila* gene have cooler CT [12,49]. Tar’an et al. [50] identified major loci associated with lodging resistance in pea on LG III, and one of the markers we identified for CT is also on LG III, suggesting genes controlling lodging also control CT. 

We identified seven MTAs associated with reproductive stem length on chromosomes 3 (LG5), 4 (LG4), 5 (LG3), and 7 (LG7); and six MTAs associated with internode length on chromosomes 4 (LG4), 5 (LG3), 6 (LG2) and 7 (LG7). The markers associated with these two traits mostly were positioned on the same linkage groups, and a SNP marker Chr5LG3_566189271 was associated with both the traits. Using the current GWAS panel, Gali et al. [30] identified four MTAs associated with plant height that were on same linkage group as that of reproductive stem length and internode length. The SNP marker Chr5LG3_566189271 reported for plant height [34] was also associated with internode length in the current study. Both reproductive stem and internode lengths were significantly reduced by heat stress [12]. A cultivar’s genetics affects internode length, and in pea the Le gene controls internode length [25], which directly affects reproductive stem length and plant height via its influence on gibberellic acid function on growth and determinacy/indeterminacy [25,51,52]. Using two pea recombinant inbred populations, Weeden [22] identified a major QTL on LG3 for a longer internode (Le), and a second QTL on LG4 for the recessive allele which caused plants to have shorter internodes. 

We identified nine loci associated with pod number, of which six were on Chr 2 (LG1), one each on Chr 3 (LG5) and 5 (LG3), and one on a non-chromosomal scaffold. Plant pod number is the number of flower-bearing nodes multiplied by the average number of flowers per node. Previously, Jiang et al. [28] identified two unmapped QTLs for pod number using 92 diverse accessions. Also, Huang et al. [24] identified two QTLs for pod number based on a bi-parental mapping population on Chr 5 (LG3). The greater number of loci identified in this study was likely due to the use of a GWAS panel which represented a broad range of diversity in pod number ranging from 3 to 19 pods per plant, and where most of this variation is contributed by genetic factors. Benlloch [53] indicated that flower number per plant, which directly determines pod number, is controlled by two genes, Fn and Fna, and a single mutation of these genes increases flower number per plant. Pod number is a major yield component that has a strong correlation with seed yield, and is most affected by heat stress [12,23,24]. The reduction in pod number and yield was likely from heat stress-induced abortion of flower buds, flowers, and pods [4,23]. Pod set relies on pollen and stigma functioning optimally, both of which are very sensitive to heat stress [54]. 

In conclusion, in this GWAS we identified 32 MTAs and 48 candidate genes for traits associated with pea heat response. These results are expected to enhance the understanding of genetic loci controlling these traits. The identified candidate genes are involved in various biological functions and require further functional validation. The detected MTAs and candidate genes should be useful for marker-assisted selection for heat tolerant pea varieties. 

## 4. Materials and Methods

### 4.1. Plant Materials 

A panel of 135 diverse field pea accessions, as described by Gali et al. [30], were grown for two years (2016–2017) at two Rosthern (52°66’N, 106°33’W; Orthic Black Chernozem); and three years (2015–2017) at Saskatoon (52°12’N, 106°63’W; Dark Brown Chernozem), Saskatchewan, in western Canada, for phenotypic evaluation. The combination of year-location forms five environments: 2015 Saskatoon; 2016 Rosthern; 2016 Saskatoon; 2017 Rosthern; and 2017 Saskatoon for phenotypic evaluation. Among the 135 accessions, 19 were from Australian pulse breeding programs, 77 were from eastern and western European countries, the Russian Federation and the UK, 15 were from the USA, 17 were from Canada (mostly from the Crop Development Centre, University of Saskatchewan), five were from Ethiopia, and two were from India. Thus, the accessions represented the major pea growing areas of the world. The accessions were commercial cultivars released over the past 50 years for local production, and were able to flower and mature under the five environments tested [30].

### 4.2. The Field Trials and Weather Conditions

The experimental design at each environment was a randomized complete block with two replications. Plot size was 1.37 m width × 3.66 m length, and the recommended seeding rate (100 seeds m^−2^, targeting 80–85 plants m^−2^ on 0.25 m row spacing) was used. Weed control was achieved by management practices used in pea production in Saskatchewan as described by Tafesse et al. [12]. 

Weather data for 2015 Saskatoon starting from June 11 to the end of the growing season, 2016 Rosthern starting from June 21 to the end of the growing season, and 2016 Saskatoon starting from July 21 to the end of the season were collected from weather stations (Coastal Environmental Systems, Seattle, WA, USA) established at each site. Weather data of 2017 and the remaining 2015 and 2016 were obtained from Environment Canada database (https://climate.weather.gc.ca) recorded by the nearest stations to the trial sites. For Saskatoon, data from central Saskatoon station, and for Rosthern the mean of data from Saskatoon international airport and Prince Albert stations were used. The daily maximum air temperatures, amount of precipitation and number of days when air temperature exceeded 28 °C during the growing seasons were used to determine the degree of stress in each environment at different growth stages. The categorization of growth stages into vegetative (germination to end of vegetative growth) and reproductive (beginning of flowering to maturity) was conducted using the phenology data reported by Gali et al. [30]. Based on the weather data, 2015 and 2017 Saskatoon had heat and drought stress conditions and the remaining three environments were generally ambient and considered control environments (Table 1). 

### 4.3. Phenotypic Measurements

Chlorophyll concentration was estimated non-destructively using a SPAD502Plus chlorophyll meter (Konica Minolta Sensing Americas Inc., USA). The SPAD value is a unitless index, calculated as the ratio of the intensity of light transmittance at red (650 nm) to infrared (940 nm) and gives a value that corresponds to the relative amount of chlorophyll present in the leaf. Hereafter, the chlorophyll concentration estimated by SPAD meter is referred to as ‘SPAD’. The SPAD readings were taken four to six times each season, and for each measurement day the mean SPAD value was calculated by the instrument from three readings taken from three plants per plot on fully expanded stipules at the second or third node counting down from the apex of a main stem.

Similarly, spectral measurement was conducted repeatedly on leaf stipules using a portable spectroradiometer PSR-1100F (Spectral Evolution Inc, Lawrence, MA, USA). This device enabled hyperspectral readings with a range of 320-1,126 nm, and 1.6 nm sampling interval, and a total of 512 discrete narrow bands. PRI was calculated from the reflectance data according to Gamon et al. [19] as:PRI = (R_531_ − R_570_)/(R_531_ + R_570_)(1)
where R is reflectance percentage and 531 and 570 are the wavelength bands in nm along the light spectrum. The PRI is used as a proxy for the xanthophyll cycle, a photosynthetic protective cycle that operates more during stress [19]. 

Canopy temperature (CT) was measured four to six times in each location in a season using a hand held infrared thermometer (Model 6110.4ZL, Everest Interscience Inc, Tucson, AZ, USA) as described by Tafesse et al. [12]. Measurements of SPAD, spectral reflectance, and CT were carried out repeatedly (four to six times in a season) during the reproductive stage, at solar noon on relatively hot days when air temperature is greater than 25 °C, and the mean value was used for analysis.

The other measurements taken at physiological maturity were: reproductive stem length (vine length from first flowering node to the tip of the main stem); internode length (determined as the ratio of reproductive stem length to reproductive node numbers); and pod number per plant (total pods counting all pods with at least one seed on the main stem). For these, each measured variable was the mean of three plants per plot sampled at random and lengths were measured in cm. 

### 4.4. Phenotype Data Analysis

Before employing analysis of variance (ANOVA), homogeneity of variances and normality of residuals were tested using checked using Levene and Shapiro-Wilk tests, respectively [55,56]. Variance components of genotype, environment, the G × E interaction, block within environment, and the residual were analyzed using the generalized linear model (GLM) and by considering all factors as random effects. Broad sense heritability (H^2^) was calculated as:H^2^= σg^2^/(σg^2^ + σge^2^/n + σ^2^/nb)(2)
where σg^2^ is the genetic variance, and σge^2^ is the variance of genotype and environment [57].

Over environments, combined ANOVA on SPAD, PRI, CT, reproductive stem length, internode length, and pod number was carried out using the Mixed procedure of SAS (Version 9.4, SAS Institute). Genotype, environment and G x E interaction were considered as fixed, and blocks as random factors. Principal component analysis (PCA) was performed with the multivariate function of Minitab (Version 19, Minitab LLC, USA) using means of traits to infer overall association among traits and genotype for the five environments. Based on significant eigenvalue (> 1), the first two principal components (PC) were selected for the minimum number of PCs to explain the greatest total variation in the data set.

### 4.5. Association Mapping

Genotyping of the 135 GWAS panel was performed by genotyping-by-sequencing (GBS, [58]), and 16,877 SNPs were reported based on a minimum read depth of five and minimum allele frequency of 0.05 [30]. The reported SNPs were used for association analysis using GAPIT (Genome Association and Prediction Integrated Tool—R package [30]) software. Association analysis for each trait was conducted using the mixed linear model (MLM). For MLM analysis, Q values were generated from structure analysis [59] and K (kinship coefficient matrix) values calculated by GAPIT and identity-by-state (IBS) methods were used. Principal co-ordinate values were used as co-variates. Although the result is not presented here, the model output of MLM was compared with the Super MLM model and the markers identified in both methods were mostly similar. The quantile-quantile (Q-Q) plots of each trait were drawn using the observed and expected log_10_P values. Marker–trait associations were selected based on *P* value (*P* ≤ 0.001) and repeated occurrence of the association in at least three of the five trials. The genes within 15 kb of the identified markers are reported as the candidate genes. The pea genome sequence reported by Kreplak et al. [40] was used for identification of candidate genes.

## Figures and Tables

**Figure 1 ijms-21-02043-f001:**
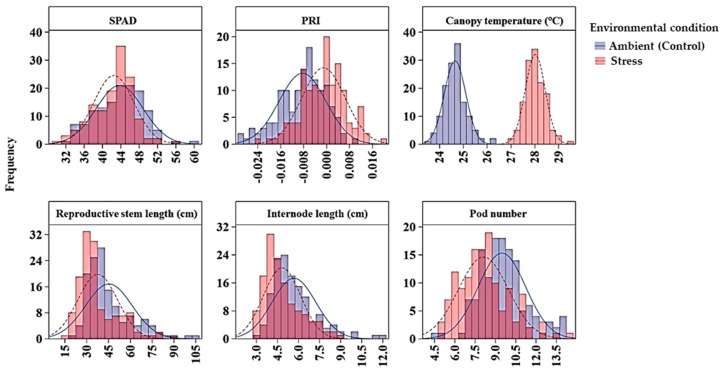
Distribution of average SPAD, PRI, canopy temperature, reproductive stem length, internode length and pod number of 135 GWAS accessions across ambient and stress environments. Note: The ambient (control) environments were 2016 Rosthern, 2016 Saskatoon and 2017 Rosthern; and the heat stress environments were 2015 and 2017 Saskatoon. PRI, photochemical reflectance index.

**Figure 2 ijms-21-02043-f002:**
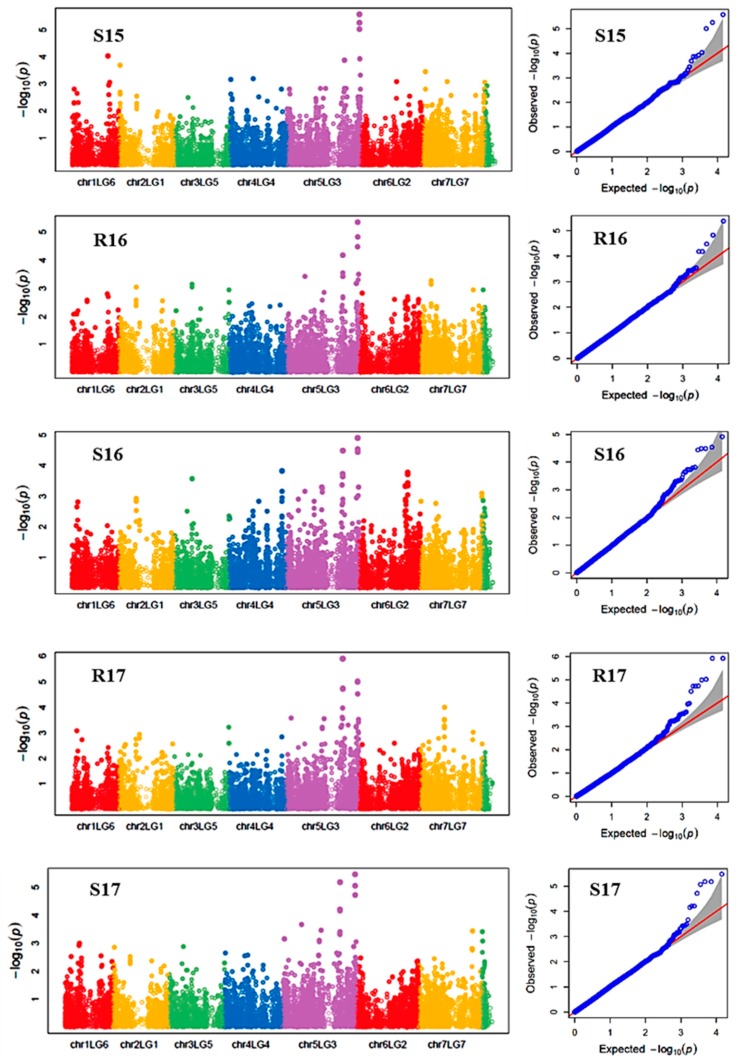
Manhattan plots and the corresponding Q-Q plots representing the identification of SNP markers associated with chlorophyll concentration measured by a SPAD meter. The Manhattan plots are based on association of 15,608 chromosomal and 1269 non-chromosomal SNPs with SPAD of 135 pea accessions in the multi-year, multi-environment trials. Note: S15, Saskatoon in 2015; R16, Rosthern in 2016; S16, Saskatoon in 2016; R17, Rosthern in 2017; and S17, Saskatoon in 2017.

**Figure 3 ijms-21-02043-f003:**
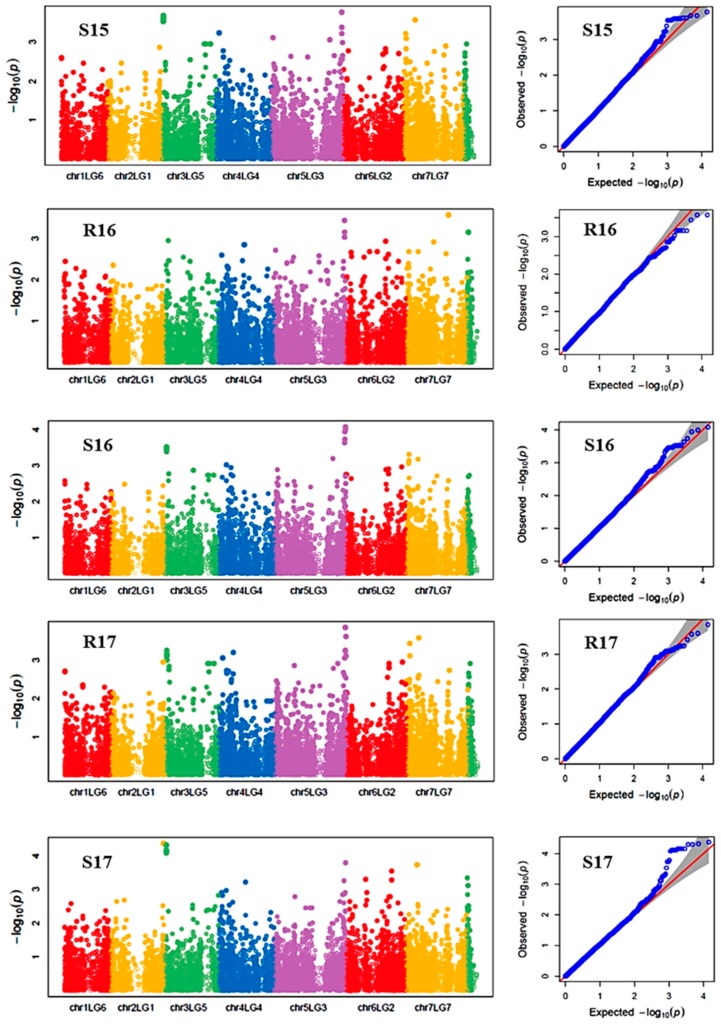
Manhattan plots and the corresponding Q-Q plots representing the identification of SNP markers associated with reproductive stem length. The Manhattan plots are based on association of 15608 chromosomal and 1269 non-chromosomal SNPs with reproductive stem length of 135 pea accessions in the multi-year, multi-environment trials. Note: R16, Rosthern in 2016; R17, Rosthern in 2017; S15, Saskatoon in 2015; S16, Saskatoon in 2016; and S17, Saskatoon in 2017.

**Figure 4 ijms-21-02043-f004:**
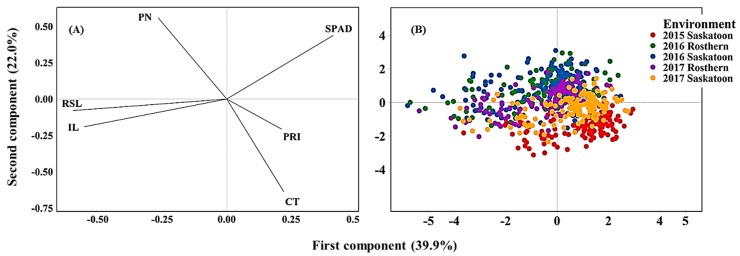
Loading (**A**) and Score (**B**) plots of principal component analysis illustrating the overall association of traits and genotype performance across environments. Note: PN, pod number; RSL, reproductive stem length; IL, internode length, CT, canopy temperature; PRI, photochemical reflectance index.

**Table 1 ijms-21-02043-t001:** Seeding date, average maximum, minimum and 24 h daily mean temperatures, number of days when the daily maximum temperature was greater than 28 °C, and total monthly precipitation at different growth and development stages of pea at each environment.

Environment	Seeding Date	Growth and Development Stage	Number of Days Spent in the Growth and Development Stage	Daily Maximum Mean Temp. (°C)	Daily Minimum Mean Temp. (°C)	Daily 24 h Mean Temp. (°C)	Number of Days when Temp. was > 28 °C	Total Precipitation (mm)	Stress Situation
2015 Saskatoon	24-Apr	Germination to late vegetative stage	58	20.4b	5.3b	13.1b	7	23.1	Drought
2016 Rosthern	06-May		46	20.8ab	6.4ab	14.4ab	3	75.8	Control
2016 Saskatoon	26-Apr		50	21.5ab	6.1ab	14.6ab	8	63.7	Control
2017 Rosthern	21-May		44	22.3ab	7.4a	15.9a	5	62.1	Control
2017 Saskatoon	30-Apr		51	22.7a	6.2ab	14.7ab	9	58.5	Control
2015 Saskatoon		Beginning of flowering to maturity	42	27.1a	14.0a	20.0a	18	41.3	Heat, drought
2016 Rosthern			52	23.1d	13.4a	18.5b	4	126.2	Control
2016 Saskatoon			48	24.4cd	11.9b	18.2b	3	86.2	Control
2017 Rosthern			46	25.6bc	10.3c	18.3b	9	46.7	Drought
2017 Saskatoon			44	25.9ab	10.4c	18.6b	16	42.6	Heat, drought

Note: temp, temperature; mm, millimeter; Means of environmental variables that do not share a letter within a column under each growth stage are significantly different from each other. The data were analyzed by one-way ANOVA and followed with the Tukey–HSD test for the mean separations.

**Table 2 ijms-21-02043-t002:** Variance components of environment, genotype, and their interaction and broad sense heritability (H^2^) on SPAD, PRI, canopy temperature, reproductive stem length, internode length and pod number in 135 pea accessions.

Source	SPAD		PRI		Canopy Temperature		Reproductive Stem Length		Internode Length		Pod Number	
	Variance	% of Total	Variance	% of Total	Variance	% of Total	Variance	% of Total	Variance	% of Total	Variance	% of Total
Genotype (G)	19.88 ***	67.9	0.0000171 ***	4.8	0.095 ***	1.7	189.12 ***	63.4	1.69 ***	43.0	2.33 ***	36.6
Environment (E)	0.64 ***	2.2	0.000067 ***	18.7	4.70 ***	85.3	22.52 ***	7.6	0.19 **	4.8	0.79 ***	12.4
REP	0.05 **	0.2	0 ns	0.0	0.006 ns	0.1	8.72	2.9	0.11 **	2.7	0.00 ns	0.0
G × E	1.47 ***	5.0	0.00041 ***	11.4	0.00 ns	0.0	7.58 **	2.5	0 ns	0.0	0.07	1.1
Error	7.25	24.7	0.000233	65.1	0.71	12.9	145	23.6	1.94	49.5	3.18	49.9
Total	29.29		0.00036		5.51		298.19		3.93		6.36	
(H^2^)	0.95		0.35		0.57		0.92		0.90		0.88	

Note: * Significant at the 0.05 level of probability; ** Significant at the 0.01 level of probability; *** Significant at the 0.001 level of probability; ns, not significant at the 0.05 level. SPAD, soil plant analysis development; PRI, photochemical reflectance index.

**Table 3 ijms-21-02043-t003:** Minimum, maximum and mean values of phenotypic traits of 135 pea accessions of the genome-wide association study panel.

Trait	Environment	Minimum	Maximum	Mean	Standard Deviation
SPAD	2015 Saskatoon	27.3	57.6	42.5	4.7
	2016 Rosthern	30.0	67.5	45.0	6.7
	2016 Saskatoon	31.0	61.1	43.7	4.8
	2017 Rosthern	32.5	56.8	42.9	5.0
	2017 Saskatoon	26.6	55.7	42.6	5.2
Photochemical reflectance index (PRI)	2015 Saskatoon	−0.039	0.028	0.000	0.012
	2016 Rosthern	−0.032	0.028	0.001	0.012
	2016 Saskatoon	−0.116	0.024	−0.019	0.024
	2017 Rosthern	−0.031	0.02	−0.006	0.01
	2017 Saskatoon	−0.037	0.026	−0.003	0.013
Canopy temperature (°C)	2015 Saskatoon	28.0	31.0	29.6	0.5
	2016 Rosthern	21.4	26.9	24.2	1.0
	2016 Saskatoon	22.3	28.4	24.6	1.2
	2017 Rosthern	23.5	26.9	25.1	0.6
	2017 Saskatoon	24.5	29.1	26.4	0.8
Reproductive stem length (cm)	2015 Saskatoon	13.2	90.7	37.9	15.0
	2016 Rosthern	16.0	117	48.9	19.7
	2016 Saskatoon	14.4	101	42.9	17.6
	2017 Rosthern	18.3	104	42.0	15.3
	2017 Saskatoon	14.6	99	36.0	15.1
Internode length (cm)	2015 Saskatoon	1.6	10.7	4.7	1.6
	2016 Rosthern	2.0	14.7	5.8	2.1
	2016 Saskatoon	1.9	14.7	5.1	2.0
	2017 Rosthern	2.4	14.9	6.0	2.0
	2017 Saskatoon	1.9	11.3	4.9	1.7
Pod number	2015 Saskatoon	3.0	13.0	7.8	1.8
	2016 Rosthern	3.5	18.5	9.8	2.8
	2016 Saskatoon	3.0	17.5	9.9	2.6
	2017 Rosthern	4.0	15.0	8.6	2.0
	2017 Saskatoon	4.5	18.5	8.3	2.4

Note: soil plant analysis development (SPAD), spectral reflectance and canopy temperature were taken four to six times in a season during reproductive stage on hot days at solar noon. A SPAD reading > 50 indicates a dark-green color and high chlorophyll concentration, a reading < 40 indicates a yellow-green color and low chlorophyll concentration. Reproductive stem length, internode length and pod number were measured on three plants per plot at physiological maturity. The overall weather classification of environments 2015 and 2017 at Saskatoon was heat stress, and the remaining three environments condition was ambient (control) for pea production. A SPAD value is an index of light transmittance at 650 nm and 940 nm. Similarly, PRI is an index derived from narrow-band (531 and 571 nm) spectral reflectance.

**Table 4 ijms-21-02043-t004:** Trait-linked SNP markers identified by association analysis of pea phenotypes associated with heat stress using the mixed linear model (MLM).

Trait	SNP Marker	Environment	*p*.value	R Square of Model with SNP	R Square of Marker ^†^	Average R Square of Marker
SPAD	Chr5LG3_150942510	2016 Rosthern	3.77 × 10^−4^	0.39	0.08	
		2016 Saskatoon	6.80 × 10^−4^	0.45	0.06	
		2017 Saskatoon	2.15 × 10^−4^	0.42	0.09	0.08
	Chr5LG3_446272814	2016 Saskatoon	1.89 × 10^−4^	0.46	0.08	
		2017 Rosthern	2.46 × 10^−4^	0.48	0.07	
		2017 Saskatoon	4.68 × 10^−4^	0.41	0.08	0.08
	Chr5LG3_449362407	2015 Saskatoon	1.39 × 10^−4^	0.42	0.09	
		2016 Rosthern	6.66 × 10^−5^	0.41	0.1	
		2016 Saskatoon	3.27 × 10^−5^	0.48	0.09	
		2017 Rosthern	1.24 × 10^−6^	0.54	0.13	
		2017 Saskatoon	6.61 × 10^−6^	0.46	0.13	0.11
	Chr5LG3_566189589	2015 Saskatoon	5.00 × 10^−7^	0.56	0.15	
		2016 Rosthern	4.33 × 10^−6^	0.45	0.14	
		2016 Saskatoon	1.23 × 10^−5^	0.49	0.1	
		2017 Rosthern	9.83 × 10^−6^	0.52	0.11	0.13
	Chr5LG3_569788697	2015 Saskatoon	1.22 × 10^−4^	0.42	0.09	
		2016 Rosthern	5.03 × 10^−4^	0.39	0.08	
		2016 Saskatoon	9.70 × 10^−4^	0.45	0.06	
		2017 Rosthern	9.00 × 10^−4^	0.47	0.06	0.07
	Chr5LG3_572899434	2015 Saskatoon	4.76 × 10^−4^	0.41	0.08	
		2016 Rosthern	3.17 × 10^−4^	0.39	0.08	
		2016 Saskatoon	5.09 × 10^−4^	0.45	0.06	
		2017 Rosthern	2.98 × 10^−4^	0.48	0.07	0.07
PRI	Chr6LG2_469101917	2016 Rosthern	8.99 × 10^−4^	0.3	0.08	
		2017 Rosthern	8.85 × 10^−5^	0.3	0.11	
		2017 Saskatoon	3.39 × 10^−3^	0.16	0.07	0.09
	Chr7LG7_263964018	2016 Rosthern	8.99 × 10^−4^	0.3	0.08	
		2017 Rosthern	8.85 × 10^−5^	0.3	0.11	
		2017 Saskatoon	3.39 × 10^−3^	0.16	0.07	0.09
Canopy temperature	Chr4LG4_415994869	2015 Saskatoon	1.16 × 10^−3^	0.52	0.05	
		2016 Rosthern	1.08 × 10^−3^	0.5	0.06	
		2016 Saskatoon	2.22 × 10^−4^	0.44	0.08	0.06
	Chr5LG3_309595819	2015 Saskatoon	4.88 × 10^−4^	0.53	0.06	
		2016 Rosthern	5.11 × 10^−3^	0.48	0.04	
		2016 Saskatoon	4.39 × 10^−4^	0.43	0.07	0.06
Reproductive stem length	Chr3LG5_18678117	2015 Saskatoon	2.18 × 10^−4^	0.63	0.06	
		2016 Saskatoon	3.60 × 10^−4^	0.62	0.05	
		2017 Rosthern	6.62 × 10^−4^	0.7	0.04	
		2017 Saskatoon	8.42 × 10^−5^	0.5	0.08	0.06
	Chr4LG4_29062302	2015 Saskatoon	5.85 × 10^−4^	0.62	0.05	
		2016 Rosthern	2.58 × 10^−3^	0.59	0.03	
		2016 Saskatoon	2.09 × 10^−3^	0.61	0.04	
		2017 Rosthern	8.96 × 10^−4^	0.7	0.03	
		2017 Saskatoon	3.11 × 10^−3^	0.46	0.04	0.04
	Chr5LG3_566189271	2015 Saskatoon	1.72 × 10^−4^	0.63	0.06	
		2016 Rosthern	3.71 × 10^−4^	0.61	0.05	
		2016 Saskatoon	1.14 × 10^−4^	0.63	0.06	
		2017 Rosthern	1.43 × 10^−4^	0.71	0.04	0.05
	Chr5LG3_572669963	2015 Saskatoon	1.06 × 10^−3^	0.62	0.05	
		2016 Saskatoon	1.03 × 10^−4^	0.63	0.06	
		2017 Rosthern	2.53 × 10^−4^	0.71	0.04	0.05
	Chr7LG7_20086906	2015 Saskatoon	6.08 × 10^−4^	0.62	0.05	
		2016 Rosthern	4.27 × 10^−3^	0.59	0.03	
		2016 Saskatoon	8.52 × 10^−4^	0.61	0.04	
		2017 Rosthern	4.00 × 10^−3^	0.69	0.03	0.04
	Chr7LG7_23295474	2015 Saskatoon	8.25 × 10^−4^	0.62	0.05	
		2016 Saskatoon	4.84 × 10^−4^	0.62	0.05	
		2017 Rosthern	3.82 × 10^−4^	0.7	0.03	0.05
	Chr7LG7_96157380	2015 Saskatoon	2.72 × 10^−4^	0.63	0.06	
		2016 Rosthern	2.15 × 10^−3^	0.59	0.04	
		2016 Saskatoon	6.82 × 10^−4^	0.62	0.05	
		2017 Rosthern	2.68 × 10^−4^	0.71	0.04	0.05
Internode length	Chr4LG4_62461234	2015 Saskatoon	8.58 × 10^−3^	0.49	0.04	
		2016 Saskatoon	3.83 × 10^−4^	0.48	0.07	
		2017 Saskatoon	3.18 × 10^−4^	0.39	0.08	0.06
	Chr4LG4_63111072	2015 Saskatoon	3.86 × 10^−4^	0.52	0.06	
		2017 Rosthern	3.54 × 10^−3^	0.62	0.04	
		2017 Saskatoon	3.68 × 10^−4^	0.39	0.08	0.06
	Chr4LG4_80759704	2016 Rosthern	3.50 × 10^−3^	0.36	0.05	
		2016 Saskatoon	2.28 × 10^−4^	0.49	0.03	
		2017 Rosthern	7.64 × 10^−3^	0.62	0.08	0.06
	Chr5LG3_566189271	2015 Saskatoon	1.22 × 10^−5^	0.55	0.09	
		2016 Rosthern	8.23 × 10^−4^	0.38	0.07	
		2016 Saskatoon	4.72 × 10^−5^	0.5	0.09	
		2017 Rosthern	2.29 × 10^−3^	0.63	0.04	
		2017 Saskatoon	2.85 × 10^−3^	0.36	0.05	0.07
	Chr6LG2_420562729	2015 Saskatoon	3.76 × 10^−4^	0.52	0.06	
		2016 Saskatoon	3.87 × 10^−3^	0.46	0.05	
		2017 Rosthern	8.96 × 10^−4^	0.63	0.04	0.05
	Chr7LG7_197862543	2015 Saskatoon	4.69 × 10^−4^	0.52	0.06	
		2016 Saskatoon	9.72 × 10^−3^	0.45	0.05	
		2017 Saskatoon	1.39 × 10^−3^	0.37	0.06	0.06
Pod number	Chr2LG1_4359822	2015 Saskatoon	8.14 × 10^−4^	0.24	0.08	
		2016 Rosthern	1.75 × 10^−3^	0.27	0.07	
		2016 Saskatoon	3.00 × 10^−3^	0.16	0.08	0.08
	Chr2LG1_105547608	2015 Saskatoon	3.98 × 10^−4^	0.25	0.09	
		2016 Saskatoon	3.01 × 10^−3^	0.16	0.08	
		2017 Saskatoon	9.05 × 10^−4^	0.22	0.09	0.09
	Chr2LG1_370541780	2015 Saskatoon	2.08 × 10^−4^	0.26	0.1	
		2016 Saskatoon	7.58 × 10^−4^	0.18	0.1	
		2017 Saskatoon	4.68 × 10^−3^	0.19	0.06	0.09
	Chr2LG1_385949935	2015 Saskatoon	3.11 × 10^−4^	0.26	0.1	
		2016 Saskatoon	8.17 × 10^−5^	0.21	0.13	
		2017 Saskatoon	1.20 × 10^−3^	0.18	0.05	0.10
	Chr2LG1_389336188	2015 Saskatoon	4.96 × 10^−4^	0.25	0.09	
		2016 Saskatoon	2.71 × 10^−3^	0.16	0.08	
		2017 Saskatoon	4.60 × 10^−4^	0.23	0.1	0.09
	Chr2LG1_402022079	2015 Saskatoon	3.58 × 10^−3^	0.22	0.06	
		2016 Rosthern	1.16 × 10^−3^	0.27	0.07	
		2016 Saskatoon	5.15 × 10^−4^	0.18	0.1	
		2016 Saskatoon	5.15 × 10^−4^	0.18	0.1	0.08
	Chr3LG5_216337201	2015 Saskatoon	4.75 × 10^−3^	0.22	0.07	
		2016 Rosthern	3.54 × 10^−3^	0.26	0.06	
		2017 Saskatoon	3.49 × 10^−4^	0.23	0.1	0.08
	Chr5LG3_530537682	2015 Saskatoon	3.32 × 10^−3^	0.22	0.06	
		2016 Rosthern	3.80 × 10^−3^	0.26	0.06	
		2016 Saskatoon	5.81 × 10^−4^	0.18	0.1	0.07
	Sc04062_32372	2015 Saskatoon	4.27 × 10^−4^	0.25	0.09	
		2016 Rosthern	8.51× 10^−3^	0.25	0.06	
		2016 Saskatoon	7.23 × 10^−3^	0.14	0.06	
		2017 Saskatoon	1.70 × 10^−5^	0.28	0.15	0.09

Note: All markers presented here were significant in at least three of five environments for a given trait. In each SNP designation, Chr and LG indicate chromosome and linkage group and the number after the _ is the base pair position. For non-chromosomal SNPs, Sc refers to scaffold followed by the scaffold number. Each locus is represented by one SNP marker of the LD block. ^†^R-square value is presented as the difference of R-square explained by the model with and without SNP.

**Table 5 ijms-21-02043-t005:** Candidate genes identified within 15 kb distance of the SNP markers identified for association with the traits of heat tolerance.

Trait^a^	SNP Marker	Gene_ID	Protein Names	Gene Names	Organism^b^	Gene Ontology IDs	Gene Ontology (GO)
SPAD	chr5LG3_446272814	*Psat5g221440*	Amidohydrolase ytcj-like protein (Fragment)	L195_g035501	Tp	GO:0016810	hydrolase activity, acting on carbon-nitrogen (but not peptide) bonds [GO:0016810]
	chr5LG3_449362407	*Psat5g224400*	cysteine-rich receptor-like protein kinase 25	LOC101505680	Ca	GO:0004672; GO:0005524; GO:0016021	integral component of membrane [GO:0016021]; ATP binding [GO:0005524]; protein kinase activity [GO:0004672]
		*Psat5g224360*	Pentatricopeptide repeat-containing protein at1g11290-like protein	L195_g006458	Tp	GO:0008270	zinc ion binding [GO:0008270]
		*Psat5g224280*	Pentatricopeptide repeat-containing protein at1g11290-like protein	L195_g022714	Tp	GO:0008270	zinc ion binding [GO:0008270]
	chr5LG3_566189589	*Psat5g299080*	Kinesin-related protein 4-like	L195_g011972	Tp		
		*Psat5g299040*	PPR containing plant-like protein (Putative tetratricopeptide-like helical domain-containing protein)	11431556 MTR_2g102210 MtrunA17_Chr2g0331911	Mt (Mtr)		
	chr5LG3_569788697	*Psat5g301440*	Embryo-specific 3 (Fragment)	L195_g051812	Tp		
		*Psat5g301400*	Nuclear pore protein	LOC101492584	Ca	GO:0005643; GO:0015031; GO:0016020; GO:0017056; GO:0051028	membrane [GO:0016020]; nuclear pore [GO:0005643]; structural constituent of nuclear pore [GO:0017056]; mRNA transport [GO:0051028]; protein transport [GO:0015031]
	chr5LG3_572899434	*Psat5g303880*	Putative sterile alpha motif/pointed domain-containing protein (SAM domain protein)	11433470 MTR_2g102140 MtrunA17_Chr2g0331871	Mt (Mtr)	GO:0045892	negative regulation of transcription, DNA-templated [GO:0045892]
		*Psat5g303840*	putative gamma-glutamylcyclotransferase At3g02910	LOC101506022	Ca	GO:0016740; GO:0061929	gamma-glutamylaminecyclotransferase activity [GO:0061929]; transferase activity [GO:0016740]
		*Psat5g303800*	protein NUCLEAR FUSION DEFECTIVE 4	LOC101504533	Ca	GO:0016021	integral component of membrane [GO:0016021]
		*Psat5g303760*	Uncharacterized protein	L195_g009520	Tp		
PRI	chr6LG2_469101917	*Psat6g234040*	Putative GTP 3′,8-cyclase (EC 4.1.99.22)	MtrunA17_Chr1g0212051	Mt (Mtr)	GO:0006777	Mo-molybdopterin cofactor biosynthetic process [GO:0006777]
		*Psat6g234000*	Riboflavin biosynthesis protein ribF	L195_g000443	Tp	GO:0003919; GO:0009231	FMN adenylyltransferase activity [GO:0003919]; riboflavin biosynthetic process [GO:0009231]
	chr7LG7_263964018	*Psat7g148080*	TATA-binding-like protein	L195_g000140	Tp	GO:0005524	ATP binding [GO:0005524]
CT	chr4LG4_415994869	*Psat4g203800*	ethylene-responsive transcription factor-like protein At4g13040	LOC105851094	Ca	GO:0003677; GO:0003700; GO:0005634	nucleus [GO:0005634]; DNA binding [GO:0003677]; DNA-binding transcription factor activity [GO:0003700]
		*Psat4g203760*	NA	NA	NA	NA	NA
	chr5LG3_309595819	*Psat5g169800*	ABC transporter C family member 3-like isoform X1	LOC101491790	Ca	GO:0005524; GO:0016021; GO:0042626	integral component of membrane [GO:0016021]; ATP binding [GO:0005524]; ATPase activity, coupled to transmembrane movement of substances [GO:0042626]
		*Psat5g169760*	Retrovirus-related Pol polyprotein from transposon TNT 1-94	KK1_037587	Cc (Ci)	GO:0000943; GO:0003676; GO:0015074	retrotransposon nucleocapsid [GO:0000943]; nucleic acid binding [GO:0003676]; DNA integration [GO:0015074]
RSL	chr3LG5_18678117	*Psat3g006600*	uncharacterized protein LOC101515092	LOC101515092	Ca	GO:0016021	integral component of membrane [GO:0016021]
		*Psat3g006560*	L-allo-threonine aldolase-like protein (Putative aldehyde-lyase) (EC 4.1.2.-)	25499717 MTR_7g115690 MtrunA17_Chr7g0274621	Mt (Mtr)	GO:0006520; GO:0016829	lyase activity [GO:0016829]; cellular amino acid metabolic process [GO:0006520]
	chr4LG4_29062302	*Psat4g020520*	Alkaline-phosphatase-like protein (Putative Type I phosphodiesterase/nucleotide pyrophosphatase/phosphate transferase)	25494146 MTR_4g123557 MtrunA17_Chr4g0069621	Mt (Mtr)	GO:0006506; GO:0016021; GO:0051377	integral component of membrane [GO:0016021]; mannose-ethanolamine phosphotransferase activity [GO:0051377]; GPI anchor biosynthetic process [GO:0006506]
	chr5LG3_566189271	*Psat5g299080*	Kinesin-related protein 4-like	L195_g011972	Tp		
		*Psat5g299040*	PPR containing plant-like protein (Putative tetratricopeptide-like helical domain-containing protein)	11431556 MTR_2g102210 MtrunA17_Chr2g0331911	Mt (Mtr)		
	chr5LG3_572669963	*Psat5g303680*	Putative sterile alpha motif/pointed domain-containing protein (SAM domain protein)	11430703 MTR_2g104230 MtrunA17_Chr2g0333351	Mt (Mtr)		
	chr7LG7_20086906	*Psat7g013080*	aldehyde dehydrogenase family 2 member C4-like	LOC101493969	Ca	GO:0016620	oxidoreductase activity, acting on the aldehyde or oxo group of donors, NAD or NADP as acceptor [GO:0016620]
		*Psat7g013040*	Cst complex subunit ctc1-like protein	L195_g004297	Tp	GO:0000723	telomere maintenance [GO:0000723]
	chr7LG7_23295474	*Psat7g015240*	Ribosomal L7Ae/L30e/S12e/Gadd45 family protein	L195_g030323	Tp		
		*Psat7g015200*	Tesmin/TSO1-like CXC domain protein	11408106 MTR_8g103320	Mt (Mtr)		
		*Psat7g015160*	NA	NA	NA	NA	NA
	chr7LG7_96157380	*Psat7g057080*	tRNA (Cytosine(34)-C(5))-methyltransferase-like protein	25501876 MTR_8g089980	Mt (Mtr)	GO:0003723; GO:0016428	RNA binding [GO:0003723]; tRNA (cytosine-5-)-methyltransferase activity [GO:0016428]
		*Psat7g057040*	tRNA (Cytosine(34)-C(5))-methyltransferase-like protein	25501876 MTR_8g089980	Mt (Mtr)	GO:0003723; GO:0016428	RNA binding [GO:0003723]; tRNA (cytosine-5-)-methyltransferase activity [GO:0016428]
IL	chr4LG4_63111072	*Psat4g039600*	Eukaryotic translation initiation factor 3 subunit C (eIF3c) (Eukaryotic translation initiation factor 3 subunit 8) (eIF3 p110)	LOC101499912	Ca	GO:0001732; GO:0003743; GO:0005852; GO:0016282; GO:0031369; GO:0033290	eukaryotic 43S preinitiation complex [GO:0016282]; eukaryotic 48S preinitiation complex [GO:0033290]; eukaryotic translation initiation factor 3 complex [GO:0005852]; translation initiation factor activity [GO:0003743]; translation initiation factor binding [GO:0031369]; formation of cytoplasmic translation initiation complex [GO:0001732]
	chr4LG4_80759704	*Psat4g047680*	NA	NA	NA	NA	NA
		*Psat4g047640*	Ras GTPase-activating protein-binding protein 1-like	L195_g006539	Tp	GO:0003723	RNA binding [GO:0003723]
		*Psat4g047600*	Uncharacterized protein	L195_g056003	Tp	GO:0005739	mitochondrion [GO:0005739]
	chr5LG3_566189271	*Psat5g299080*	Kinesin-related protein 4-like	L195_g011972	Tp		
		*Psat5g299040*	PPR containing plant-like protein (Putative tetratricopeptide-like helical domain-containing protein)	11431556 MTR_2g102210 MtrunA17_Chr2g0331911	Mt (Mtr)		
	chr6LG2_420562729	*Psat6g211160*	Transmembrane amino acid transporter family protein	25485307 MTR_1g105980	Mt (Mtr)	GO:0016021	integral component of membrane [GO:0016021]
	chr7LG7_197862543	*Psat7g120120*	Uncharacterized protein	11443456 MTR_4g087360 MtrunA17_Chr4g0045601	Mt (Mtr)		
PN	chr2LG1_105547608	*Psat2g060680*	Uncharacterized protein	L195_g033306	Tp	GO:0003676; GO:0008270	nucleic acid binding [GO:0003676]; zinc ion binding [GO:0008270]
	chr2LG1_370541780	*Psat2g144160*	Pectin acetylesterase (EC 3.1.1.-)	LOC101497691	Ca	GO:0005576; GO:0005618; GO:0016021; GO:0016787; GO:0071555	cell wall [GO:0005618]; extracellular region [GO:0005576]; integral component of membrane [GO:0016021]; hydrolase activity [GO:0016787]; cell wall organization [GO:0071555]
	chr2LG1_385949935	*Psat2g155280*	60S ribosomal protein l8-like	L195_g013966	Tp	GO:0003735; GO:0005840; GO:0006412	ribosome [GO:0005840]; structural constituent of ribosome [GO:0003735]; translation [GO:0006412]
	chr2LG1_389336188	*Psat2g157440*	Putative ATPase, AAA-type, core, AAA-type ATPase domain-containing protein (p-loop nucleoside triphosphate hydrolase superfamily protein)	11412855 MTR_5g020990 MtrunA17_Chr5g0404661	Mt (Mtr)	GO:0005524; GO:0016787	ATP binding [GO:0005524]; hydrolase activity [GO:0016787]
	chr2LG1_402022079	*Psat2g166600*	probable serine/threonine-protein kinase At1g01540 isoform X1	LOC101489894	Ca	GO:0004672; GO:0005524; GO:0016021	integral component of membrane [GO:0016021]; ATP binding [GO:0005524]; protein kinase activity [GO:0004672]
		*Psat2g166560*	PI-PLC X domain-containing protein At5g67130	LOC101489369	Ca	GO:0006629; GO:0008081	phosphoric diester hydrolase activity [GO:0008081]; lipid metabolic process [GO:0006629]
		*Psat2g166520*	Putative rapid ALkalinization Factor (RALF)	11409897 MTR_5g017160 MtrunA17_Chr5g0402121	Mt (Mtr)		
	chr2LG1_4359822	*Psat2g005000*	Nup133/Nup155-like nucleoporin	11434873 MTR_5g097260	Mt (Mtr)	GO:0005623; GO:0017056	cell [GO:0005623]; structural constituent of nuclear pore [GO:0017056]
		*Psat2g004960*	Cation-transporting ATPase plant (Putative calcium-transporting ATPase) (EC 3.6.3.8)	11434874 MTR_5g097270 MtrunA17_Chr5g0447521	Mt (Mtr)	GO:0000166; GO:0016021	integral component of membrane [GO:0016021]; nucleotide binding [GO:0000166]
	chr3LG5_216337201	*Psat3g111000*	Phosphomannomutase (EC 5.4.2.8)	11436930 MTR_7g076670	Mt (Mtr)	GO:0004615; GO:0005737; GO:0009298	cytoplasm [GO:0005737]; phosphomannomutase activity [GO:0004615]; GDP-mannose biosynthetic process [GO:0009298]
		*Psat3g110960*	bifunctional protein FolD 4, chloroplastic	LOC101496397	Ca	GO:0004488	methylenetetrahydrofolate dehydrogenase (NADP+) activity [GO:0004488]
	chr5LG3_530537682	*Psat5g270480*	Heat shock protein 70 (HSP70)-interacting protein, putative	25487616 MTR_2g090135	Mt (Mtr)		

Note: The pea genome sequence reported by Kreplak et al. [40] was used for identification of candidate genes. The reported gene annotation and nomenclature was followed. ^a^ SPAD, soil plant analysis development; PRI, photochemical reflectance index; CT, canopy temperature; RSL, reproductive stem length; IL, internode length; PN, pod number; ^b^ Tp*, Trifolium pratense* (Red clover); Ca, *Cicer arietinum* (Chickpea) (Garbanzo); Mt, *Medicago truncatula* (Barrel medic); Mtr, *Medicago tribuloides*; Cc, *Cajanus cajan* (Pigeon pea); Ci, *Cajanus indicus*.

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
