# Peer review of "Genome-Wide Association Mapping for Heat Stress Responsive Traits in Field Pea"

_ijms, 2020, doi:10.3390/ijms21062043_

Round 1
Reviewer 1 Report
The study provided some markers associated with plant physiological and morphological traits in five environments differing in meteorological conditions. Authors synonymize these traits with heat tolerance (even in the abstract) , however, these traits are responsive to numerous environmental cues including, among others, temperature, precipitation and insolation. Heat tolerance and heat responsiveness are big terms which require solid justification and several independent lines of evidence.
The study is aiming at heat stress response but no single meteorological parameter was measured at any of the two locations during the study. All conclusions on heat responsiveness are based on extrapolation of data from stations located at physical distance similar or bigger than the distance between these two locations. In Canada there is strong North-South temperature gradient, and these locations follow this gradient, therefore presented extrapolation is prone to big errors. The lack of real environmental data is a major drawback of this study.
The emphasis could be placed rather on environmental variables hypothetically differing in these environment than on the heat itself. Indeed, discrimination between normal and heat stress conditions should be based on comparison of conditions which occurred during selected period of days from the sowing date, not from the first day of May (or any random date from the calendar). Data provided in Table 1 are misleading, because there are calendar month averages but the difference in sowing date is as big as … 4 weeks, i.e. one month without two days. It makes this table completely useless. To enable comparison between environments, Authors should provide a table with meteorological conditions averaged/summarized (whatever applicable) for every 30 days starting exactly from the sowing date in the particular site/year, ie. every 30 days from April 24th for Saskatoon 2015, every 30 days for May 21 for Rosthern 2017 etc. Then, statistical analysis should be performed separately for every 30-days period and values significantly differing from the others should be indicated - separately for mean maximum temperature, mean minimum temperature, mean 24 h temperature, number of days with temperature above 28 degrees Celsius, and total precipitation. Such an analysis may putatively highlight important differences not only in the number of 28-degree days but also in 30-days precipitation between locations and putative contribution of water shortage during early phenology in some environments, undermining justification for the use “heat stress responsive” term in the whole article. “Environmental-responsive” could be used instead, providing a short paragraph describing which parameters were significantly different in particular environments during main plant phenology phases, listing heat as one of them.
Heat tolerance term is used in this manuscript, however, no information is provided which lines are heat tolerant and which are heat susceptibe. Perhaps the use of heat tolerance in this study is an exaggeration, because in my opinion there is no single legit evidence for heat tolerance provided in this manuscript. I see just strong evidence for environmental response including temperature and rainfall.
Based on other published studies and ontology databases candidate genes could be annotated for their putative biological functions related to the studied traits. I understand that a particular gene may perform many functions but it is common approach in GWAS papers to screen the literature data for particular functions of candidate genes logically matching Authors’ current understanding of biological processes studied. It would be an important added value to automatic annotations published together with genome assembly. These candidate biological functions could be provided in Table 5 as well as hypothesized in discussion.
Other minor comments
Line 97 There is something missing in the last sentence of Introduction ending at “six heat responsive …”.
Line 360 How far the SGRL gene is from these MTAs on chromosome 5?
Line 114 Capitalization scheme should be coherent in all tables.
Line 248 Table 4 could be divided into few tables, one accommodating three physiological traits (SPAD, PRI and CT) and remaining separate tables for morphological traits (stem/node lengths) and pod number. If done is this way, every table will fit on one page after some editorial amendments.
Line 254 The choice of 15 kb window for candidate gene search should be explained. Does it represent average linkage disequilibrium decay, marker density or just random selection?
Line 386 However, there was a ‘2018 study on canopy temperature in a warm season legume, soybean, and some candidate genes were found and discussed there...
Line 433 Indeed, these “various biological functions” which were never provided may contain some function/s related to the traits observed in this study and as such should be provided in Table 5 and discussed here.
Line 556 Abbreviations which are used in main text (except those appearing on figures or in tables) should be used at every occurrence of a particular term. To exemplify, canopy temperature appears 20 times in the manuscript, even more frequently than the proposed abbreviation CT (15 times).
Phenotypic observations for all lines and environments, annotated according to MIAPPE recommendations (https://www.miappe.org/) should be provided as supplementary file to allow reconstruction of this analysis using published data.
Author Response
We appreciate the careful reviews we received which have helped to improve the manuscript. Following are our detailed responses to the reviewers’ recommendations. Our responses are in italic font. Sincerely, Tom Warkentin, on behalf of all co-authors
Reviewer 1
The study provided some markers associated with plant physiological and morphological traits in five environments differing in meteorological conditions. Authors synonymize these traits with heat tolerance (even in the abstract), however, these traits are responsive to numerous environmental cues including, among others, temperature, precipitation and insolation. Heat tolerance and heat responsiveness are big terms which require solid justification and several independent lines of evidence
The study is aiming at heat stress response but no single meteorological parameter was measured at any of the two locations during the study. All conclusions on heat responsiveness are based on extrapolation of data from stations located at physical distance similar or bigger than the distance between these two locations. In Canada there is strong North-South temperature gradient, and these locations follow this gradient, therefore presented extrapolation is prone to big errors. The lack of real environmental data is a major drawback of this study.
Reply: While weather data from nearby stations are useful and sufficiently represent the study sites, we also had weather stations established directly at the study sites for some of the environments. Based on the comment given here, we decided to substitute some of the weather data presented in the manuscript with the weather data collected from weather stations (Coastal Environmental Systems, Seattle, WA, USA) established at each site. However, as our weather data from onsite stations were not complete, we amalgamated data from nearby stations for the durations/environments in which we were missing data from our onsite stations. For the three environments in 2015 and 2016 we collected sufficient onsite data starting from about three weeks after seeding. In pea heat/drought stress is severe when it occurs in late vegetative or reproductive stages, and specifically our onsite weather data was available in these stages as follows: for 2015 Saskatoon starting from June 11 to the end of the growing season; for 2016 Rosthern starting from June 21 to the end of the growing season; and for 2016 Saskatoon starting from July 21 to the end of the season. For 2017 at both sites and the remaining durations of 2015 and 2016, weather data were obtained from Environment Canada database (https://climate.weather.gc.ca) recorded by the nearest stations to the trial sites. WE are hopeful that this information should satisfy the reviewer’s concern about ‘lack of real environmental data is a major drawback of this study’.
The emphasis could be placed rather on environmental variables hypothetically differing in these environment than on the heat itself. Indeed, discrimination between normal and heat stress conditions should be based on comparison of conditions which occurred during selected period of days from the sowing date, not from the first day of May (or any random date from the calendar). Data provided in Table 1 are misleading, because there are calendar month averages but the difference in sowing date is as big as … 4 weeks, i.e. one month without two days. It makes this table completely useless. To enable comparison between environments, Authors should provide a table with meteorological conditions averaged/summarized (whatever applicable) for every 30 days starting exactly from the sowing date in the particular site/year, ie. every 30 days from April 24th for Saskatoon 2015, every 30 days for May 21 for Rosthern 2017 etc. Then, statistical analysis should be performed separately for every 30-days period and values significantly differing from the others should be indicated - separately for mean maximum temperature, mean minimum temperature, mean 24 h temperature, number of days with temperature above 28 degrees Celsius, and total precipitation. Such an analysis may putatively highlight important differences not only in the number of 28-degree days but also in 30-days precipitation between locations and putative contribution of water shortage during early phenology in some environments, undermining justification for the use “heat stress responsive” term in the whole article. “Environmental-responsive” could be used instead, providing a short paragraph describing which parameters were significantly different in particular environments during main plant phenology phases, listing heat as one of them.
Reply: In Saskatchewan, Canada, the pea growing season is typically between late April to late August. We have accepted the reviewer’s recommendation to compare environmental conditions which occurred during selected periods of days from the sowing date to maturity instead of presenting calendar month conditions.However, instead of comparing the first 30 days after planting at a given location, then the next 30 days, etc, we decided to make our categorization according to the crop growth and development stages which we categorized into two major stages: 1) from germination to the end of vegetative growth, and 2) from beginning of flowering to maturity. To determine the number of days plants spent at each of the two stages, days to flowering and days to maturity data were used from our previously published paper on the same GWAS panel (Gali et al. 2019). For example, for 2015 Saksatoon, the average days to 50% flowering was 58 days and we know 50% flowering occurs approximately 10 days from the end of vegetative growth, and thus the number of days from germination to end of the vegetative stage was 48; and number of days from beginning of flowering to maturity was 52, i.e., from 49 to 100. Similarly, for 2016 Rosthern, average days to 50% flowering was reported to be 56 days and thus from day 1 to day 46 was regarded as germination and vegetative growth stage, and from day 47 to 98 was regarded as flowering to maturity stage. The same method was used for the remaining environments, and then statistical analysis was conducted to compare the five environments at each stage for degree of stress, using variables mentioned in Table 5. Based on this analysis, we have strong evidence that 2015 Saskatoon and 2017 Saskatoon were under stress conditions, and the remaining three environments were categorized as generally ambient.
Heat tolerance term is used in this manuscript, however, no information is provided which lines are heat tolerant and which are heat susceptibe. Perhaps the use of heat tolerance in this study is an exaggeration, because in my opinion there is no single legit evidence for heat tolerance provided in this manuscript. I see just strong evidence for environmental response including temperature and rainfall.
Reply: In a separate study both in field and controlled environments (Tafesse, 2018 [14]; Tafesse et al., 2019[12]), we have strong evidence to show that the traits presented here respond to heat stress. We have varieties that show a range of expression with respect to the traits we evaluated. Additionally, subsets of the GWAS varieties were classified as heat tolerant and heat susceptible in Tafesse et al. 2019 [12]. Chlorophyll content and vegetation indices like PRI respond to heat stress, and cultivars with these traits, along with their heat tolerance index, was also reported by Tafesse, 2018 [14].
In addition, we have adjusted the text in many places in this manuscript to reflect ‘heat and drought stress’ or ‘environmental stress’ or simply ‘stress’, instead of only referring to ‘heat stress’.
Based on other published studies and ontology databases candidate genes could be annotated for their putative biological functions related to the studied traits. I understand that a particular gene may perform many functions but it is common approach in GWAS papers to screen the literature data for particular functions of candidate genes logically matching Authors’ current understanding of biological processes studied. It would be an important added value to automatic annotations published together with genome assembly. These candidate biological functions could be provided in Table 5 as well as hypothesized in discussion.
Reply: Table 5 is modified and provided in accordance with the suggestions
Other minor comments
Line 97 There is something missing in the last sentence of Introduction ending at “six heat responsive …”.
Reply: Yes, the word ‘trait’ was missing and now included to make the sentence complete.
Line 360 How far the SGRL gene is from these MTAs on chromosome 5?
Reply: The following sentence has been added at line 365: ‘The SGRL gene sequence (https://www.ncbi.nlm.nih.gov/nuccore/LN810021) location in the pea genome assembly spanned between the base pair positions Chr5LG3_151800929 and Chr5LG3_151804253, and is within close proximity (858 Kbp) of the SPAD associated marker ChrLG3_150942510’.
Line 114 Capitalization scheme should be coherent in all tables.
Reply: All tables now have appropriate capitalization. The issue spotted at line 114 ‘Daily Maximum mean Temp’, is changed to ‘Daily maximum temp.’ and is consistent with other tables.
Line 248 Table 4 could be divided into few tables, one accommodating three physiological traits (SPAD, PRI and CT) and remaining separate tables for morphological traits (stem/node lengths) and pod number. If done is this way, every table will fit on one page after some editorial amendments
Reply: While we appreciate the suggestion, we are afraid splitting large tables into two or more will significantly increase the number of tables and the overall size of the manuscript. Tables 4 and 5 provide data for a given theme each, and are better kept intact.
Line 254 The choice of 15 kb window for candidate gene search should be explained. Does it represent average linkage disequilibrium decay, marker density or just random selection?
Reply: In the linkage disequilibrium analysis of this GWAS panel, we earlier reported that the average LD1/2max,90 (LD has decayed to half of r2 max,90) of seven chromosomes was 50 kb, while this distance is 20 kb for two of the chromosomes (Gali et al., 2019 [30]). Based on the LD measurement and number of genes in the proximity of identified SNP markers, we have used a 30 kb window to search for potential candidate genes.
Line 386 However, there was a ‘2018 study on canopy temperature in a warm season legume, soybean, and some candidate genes were found and discussed there
Reply: Indeed, there was a report on soybean that identified candidate genes for canopy temperature, however, to the best of our knowledge, we have not noted any papers on markers or candidate genes for canopy temperature in pea, or other cool season pulse crops.
Line 433 Indeed, these “various biological functions” which were never provided may contain some function/s related to the traits observed in this study and as such should be provided in Table 5 and discussed here.
Reply: Table 5 has been modified to address this concern.
Line 556 Abbreviations which are used in main text (except those appearing on figures or in tables) should be used at every occurrence of a particular term. To exemplify, canopy temperature appears 20 times in the manuscript, even more frequently than the proposed abbreviation CT (15 times).
Reply: The abbreviation list is now revised and the abbreviations which are in the list are now used in the manuscript body, unless it appears at the beginning of a sentence.
Phenotypic observations for all lines and environments, annotated according to MIAPPE recommendations (https://www.miappe.org/) should be provided as supplementary file to allow reconstruction of this analysis using published data.
Reviewer 2 Report
Congratulations to the authors for such a nice study. I have only a few suggestions.
Line# 30: What is a field pea?
Cut down the introduction to keep the relevant part only.
Line# 105: What do you mean by “Accordingly, environments 2015 Saskatoon and 2017 Saskatoon..”?
Author Response
We appreciate the careful reviews we received which have helped to improve the manuscript. Following are our detailed responses to the reviewers’ recommendations. Our responses are in italic font. Sincerely, Tom Warkentin, on behalf of all co-authors
Reviewer 2
Congratulations to the authors for such a nice study. I have only a few suggestions.
Line# 30: What is a field pea?
Reply: The word ‘field’ has been deleted. ‘Field pea’ is a term sometimes used to differentiate dry pea from vegetable pea, but in general, the word ‘pea’ is sufficient in reference to dry pea in the context of this paper.
Cut down the introduction to keep the relevant part only.
Reply: In our opinion, the Introduction section is already quite brief and is needed to describe the problem, the key background information, and the objectives.
Line# 105: What do you mean by “Accordingly, environments 2015 Saskatoon and 2017 Saskatoon..”?
Reply: This sentence and the following sentences were clarified as follows.
Saskatoon 2015 was the most heat stressed environment as indicated by mean daily maximum air temperatures > 27 °C; 18 days the air temperature was > 28 ℃, and it had relatively dry conditions during the reproductive stage. Similarly, 2017 Saskatoon was also under heat and drought stress during the reproductive stage with average air temperature ~ 26 °C, 16 days air temperature > 28 ℃, and relatively low total precipitation. The remaining three environments were generally ambient and considered as control environments (Table 1).
Reviewer 3 Report
The MS reports on a genome-wide association study (GWAS) for heat stress response traits in pea. Global warming has driven the subject of heat stress response to the spotlight to cold climate species (as warm climate ones have been studied for decades regarding these traits…). The subject of finding specific markers to assess a variety’s degree of tolerance is no doubt a very pertinent one and of particular interest for research in abiotic stress in crops.
Regarding this MS, I would like to see some concerns addressed by the authors:
Line 91: the sentence is incomplete.
Paragraph 1 of results: is 28º the maximum cardinal temperature? This section should be discussed according to those parameters, Tb, TB and Toptimal, as those are the temps that affect growth.
Don’t introduce Table 5 like that… if you want to indicate some more relevant genes, indicate their correct names and not the generic ones. I don´t know the gene “encoding for transcription factor”, there are hundreds of those and they are very specific so if you want to say something about it, say the actual name, otherwise don’t mention… The same for “heat shock protein”, some don’t even respond to heat shock…
Data on table 5? Please provide a NCBI database correspondence to your gene IDs.
Discussion: As you have such high phenotypic variation it would be interesting to assess the Physiological meaning of the correspondences between PRI and the genes that seem to be affecting this parameter. You give the example of Violaxanthin de-epoxidase (VDE) in your MS but you didn’t actually find it in this work… Again, for temperature indexes, you relate several genes to this trait but none of those did you describe in your work.
Line 378: …the genes known to be involved in energy…
M&M: 4.5 Association Mapping: provide a more detailed description of the analysis, not just a 9 year old generic reference…
Please provide the correct title for reference 58.
Author Response
We appreciate the careful reviews we received which have helped to improve the manuscript. Following are our detailed responses to the reviewers’ recommendations. Our responses are in italic font. Sincerely, Tom Warkentin, on behalf of all co-authors
Reviewer 2
The MS reports on a genome-wide association study (GWAS) for heat stress response traits in pea. Global warming has driven the subject of heat stress response to the spotlight to cold climate species (as warm climate ones have been studied for decades regarding these traits…). The subject of finding specific markers to assess a variety’s degree of tolerance is no doubt a very pertinent one and of particular interest for research in abiotic stress in crops.
Regarding this MS, I would like to see some concerns addressed by the authors:
Line 91: the sentence is incomplete.
Reply: The sentence has been improved as follows.
Stress tolerance is complex and is controlled by many genes throughout the genome each with minor effects and each interacting with the environment [39].
Paragraph 1 of results: is 28º the maximum cardinal temperature? This section should be discussed according to those parameters, Tb, TB and Toptimal, as those are the temps that affect growth.
Reply: Cardinal temperature determines the rate of photosynthesis and plant growth, however, the influence of cardinal temperature was not part of this study. We indicated 28oC as the threshold temperature based on previously reported threshold temperatures for pea in similar environments. When the air temperature exceeds 28oC repeatedly in a season, particularly in the reproductive stage, yield is significantly reduced. Thus, we used this temperature as a cutoff point along with the number of days the air temperature was beyond this temperature to indicate the degree of heat stress and categorize the environments into stressed and ambient.
Don’t introduce Table 5 like that… if you want to indicate some more relevant genes, indicate their correct names and not the generic ones. I don´t know the gene “encoding for transcription factor”, there are hundreds of those and they are very specific so if you want to say something about it, say the actual name, otherwise don’t mention… The same for “heat shock protein”, some don’t even respond to heat shock…
Data on table 5? Please provide a NCBI database correspondence to your gene IDs.
Reply: Table 5 has been improved to include gene names, organism, gene ontology IDs, and gene ontology (GO).
Discussion: As you have such high phenotypic variation it would be interesting to assess the Physiological meaning of the correspondences between PRI and the genes that seem to be affecting this parameter. You give the example of Violaxanthin de-epoxidase (VDE) in your MS but you didn’t actually find it in this work… Again, for temperature indexes, you relate several genes to this trait but none of those did you describe in your work.
Reply: The genes associated with the traits PRI, Putative GTP 3',8-cyclase, Riboflavin biosynthesis protein ribF, and TATA-binding-like protein are essential in numerous plant metabolic processes including in the oxidation-reduction pathways. Similarly genes associated with canopy temperature including ethylene-responsive transcription factor-like protein , ABC transporter C family member 3-like isoform X1, and retrovirus-related Pol polyprotein from transposon TNT 1-94 are involved in numerous plant metabolic processes including in stress responses.
Line 378: …the genes known to be involved in energy…
Reply: The sentence has been improved as follows: Violaxanthin de-epoxidase VDE is among the genes known to be involved in excess energy dissipation in the xanthophyll cycle [45].
M&M: 4.5 Association Mapping: provide a more detailed description of the analysis, not just a 9 year old generic reference…
Reply: More detailed description is provided along with the appropriate references. Gali et al., (2019) conducted a GWAS study on the same pea panel, the details of the association mapping method is the same as in this manuscript, and thus we referred to Gali et al., (2019) for further details of the association mapping methodology.
Please provide the correct title for reference 58.
Reply: The correct reference title is now provided as: A reference genome for pea provides insight into legume genome evolution.
Round 2
Reviewer 1 Report
The manuscript has been improved as requested. I have no further concerns.